# Early Microvascular Dysfunction: Is the Vasa Vasorum a “Missing Link” in Insulin Resistance and Atherosclerosis

**DOI:** 10.3390/ijms22147574

**Published:** 2021-07-15

**Authors:** Jeanette Owusu, Eugene Barrett

**Affiliations:** 1Department of Medicine, School of Medicine, University of Virginia, Charlottesville, VA 22908, USA; jo5fp@hscmail.mcc.virginia.edu; 2Department of Pediatrics, School of Medicine, University of Virginia, Charlottesville, VA 22908, USA; 3Department of Pharmacology, School of Medicine, University of Virginia, Charlottesville, VA 22908, USA

**Keywords:** atherosclerosis, adventitial vasa vasorum, microvascular insulin resistance, angiogenesis

## Abstract

The arterial vasa vasorum is a specialized microvasculature that provides critical perfusion required for the health of the arterial wall, and is increasingly recognized to play a central role in atherogenesis. Cardio-metabolic disease (CMD) (including hypertension, metabolic syndrome, obesity, diabetes, and pre-diabetes) is associated with insulin resistance, and characteristically injures the microvasculature in multiple tissues, (e.g., the eye, kidney, muscle, and heart). CMD also increases the risk for atherosclerotic vascular disease. Despite this, the impact of CMD on vasa vasorum structure and function has been little studied. Here we review emerging information on the early impact of CMD on the microvasculature in multiple tissues and consider the potential impact on atherosclerosis development and progression, if vasa vasorum is similarly affected.

## 1. Introduction

The vasa vasorum is the nexus of the micro- and macro-vasculature and as such forms a nidus where dysfunction of, or injury to, either element may impact the integrity of the other, setting off a chain reaction that could accelerate vascular disease development. It is both curious and unfortunate that for cardio-metabolic disease (CMD), which adversely affects both the micro- and macro-vasculature, the involvement of vasa vasorum has been little studied. In contrast, the impact of insulin resistance, hypertension, and diabetes on microvascular structure and function has been extensively studied in multiple tissues other than the arterial wall. We focus this brief review on the potential for significant parallels between findings in the vasa vasorum and microvasculature at other tissue sites. We also briefly review data indicating a central role for vasa vasorum in the pathogenesis of atherosclerotic lesions and the acceleration of that process by insulin resistance and diabetes. 

## 2. Structure and Function of the Arterial Vasa Vasorum

Before addressing vasa vasorum pathophysiology, we briefly review its normal structure and function. The arterial vasa vasorum refers to the network of arterioles, capillaries and small venules that enter and supply the arterial wall from either the luminal (vasa vasorum interna) or (much more abundantly) from the adventitial (vasa vasorum externa, or adventitial vasa vasorum) surface (Figure 1). The tunica intima of larger arteries is a thin band comprised of the endothelium with its basement membrane, occasional pericytes and adjacent connective tissue that encircles the lumen and bounded by the internal elastic lamina. It is nourished directly by diffusion of oxygen and nutrients from within the arterial lumen. Within the tunica media of the vessel wall, elastin and collagen elements as well as smooth muscle cells (SMC) are arranged in an orderly, stacked, lamellar pattern. Generally, arteries with a lumen diameter < 0.5 mm, or a wall thickness < 29 medial lamellar units [1,2] have no vasa vasorum. The media of these smaller arterial vessels, like the intima of larger vessels, receives oxygen and nutrient exchange directly from the arterial lumen. For the media of large arterial vessels, the blood carried through the vasa vasorum, arising from the adventitia, provides gas and nutrient exchange with the cellular components, mostly SMC, fibroblasts, and sympathetic nerve processes. The adventitial layer of the vasa vasorum extends outward from the external elastic membrane of the vessel wall and includes connective tissue elements (fibroblasts, adipocytes, and pre-adipocytes, elastin, collagen), as well as macrophages and leukocytes. The vasa vasorum externa arises off branches of parent arteries, investing first the arterial adventitia and proceeding inward to the arterial media.

In humans, the vasa vasorum is present in the walls of many larger arterial vessels including the coronary arteries, the aorta and its branches, (subclavian, carotid, iliac, femoral and brachial) and the intercostal arteries branching off the descending thoracic aorta [3]. Anatomically, there is the first order vasa which runs longitudinally between the adventitia and media paralleling the lumen while second order vasa run circumferentially around the vessel and often penetrates into the tunica media. 

Small laboratory rodents (rats, mice) normally lack vasa vasorum in their arterial vessels [3], however, vascular injury or atherogenesis in genetically prone laboratory rodents each can promote formation of a reactive, more inflammatory “neovascular” vasa vasorum at sites of pathologic injury. Development of this usually begins from vessels in the adventitial layer which, as noted, includes not just vessels but resident macrophages, mast cells, B and T-lymphocytes, adipocytes, fibroblasts, and progenitor cells. In non-diseased porcine coronary arteries first order vasa are more abundant than second order, while the reverse is seen in atherosclerotic models. The development of this neovascular vasa vasorum has been intensively studied and will be discussed later when considering atherogenesis. 

Along with the arterial components of the vasa vasorum there are venous vessels in the arterial wall which drain a network of venules and capillaries near its arteriolar counterpart. Separate from the arterial vasa vasorum, the walls of large veins have their own vasa vasorum. For purposes of this review, we focus solely on the arterial wall vasa vasorum and its role in normal physiology and in pathological transformation of vessels in insulin resistant states such as diabetes, pre-diabetes, and metabolic syndrome. 

The arterial vasa vasorum supplies oxygen and nutrients and removes waste products from sites in the arterial wall that cannot be adequately served by simple diffusion from blood flowing through the parent artery [4]. The essential nature of adventitial vasa vasorum was demonstrated by early studies showing that ligation of intercostal arteries (from which portions of the aortic vasa vasorum originate) produced medial necrosis of the aortic wall [5,6]. Subsequent studies by Heistad and colleagues used radiolabeled microspheres to quantify the perfusion of both canine, and primate aortic or coronary vasa vasorum [7,8]. In the outermost and middle third of the aortic wall, vasa vasorum flow averaged ~10 mL/min/100 g tissue while flow to the inner third was only 10% of this. Presumably, diffusion of oxygen and nutrients from the parent vessel lumen provided for any remaining needs. The same group of investigators demonstrated that, like microvasculature elsewhere in the body, the aortic vasa vasorum blood flow responded to sympathetic nerve and baroreceptor stimulation directly. Vasa vasorum flow is impacted by muscle tone of the aortic wall as well [8]. It is evident that, as for the myocardial microvasculature, capillary perfusion of the vasa vasorum will be limited during arterial systole when arterial wall tension is high. Indeed, vasa vasorum perfusion is limited even during diastole by diastolic wall tension which exceeds typical systemic capillary luminal pressure [1]. To our knowledge, measurements of intraluminal pressure within the distal elements of the vasa vasorum are not available, but must exceed diastolic wall pressures of the host artery for flow to occur. The wall tension in the aortic wall is greatest in the innermost third of the vessel and decreases as radial distance from the lumen increases to the point where it is not a factor for vasa vasorum perfusion in the outer adventitial layers.

Flow through the vasa vasorum responds to sympathetic tone as well as to circulating vasoactive signals including vasodilators adenosine, bradykinin, ATP and nitric oxide and vasoconstrictors including endothelin-1, and norepinephrine [4]. To our knowledge myogenic tone has not been measured, but presumably is present in the arterial vasa vasorum. Just how circulating concentrations or paracrine release of vasoactive factors integrates the day-to-day regulation of vasa vasorum perfusion is not resolved [4]. Importantly for the current discussion, whether states of insulin resistance (obesity, diabetes, metabolic syndrome, hypertension) impact vasa vasorum perfusion is not known. 

## 3. Normal Growth of the Vasa Vasorum

Anatomical data regarding the early development of the human vasa vasorum is sparse due in part to the small size and fine structure of these vessels. Careful micro-CT studies of young pigs identified vasa vasorum in 1- and 6-month-old animals and documented the vasa vasorum growth that accompanied enlargement of the parent vessels [9]. This suggests that at birth, the arterial wall of larger arteries already includes at least an adventitial vasa vasorum. The adventitia is composed of a mixture of fibroblasts, adipocytes, resident macrophages, precursor cells, and leukocytes, as well as neural elements. As growth occurs, the thickness of each of the three layers of the vessel wall increases. To meet the needs of oxygen and nutrient delivery and waste clearance, the microvascular vasa vasorum develops pari passu with the enlarging vessels. New microvascular vessels arise within the adventitia and orient longitudinally along the parent vessel with subsequent branching in a circumferential fashion and penetrating into the tunica media. Much less abundant are the vasa vasorum interna that arise from the lumen of the parent vessel. This angiogenic process during normal growth is slowly progressive, orderly, and furnishes the vessel wall with the microvascular network needed for its normal physiologic function. Normal growth of the adventitial vasa vasorum is stimulated by angiogenic factors (e.g., hypoxia-inducible transcription factors (HIF-1 and HIF-2) as well as FGF2 and VEGF) expressed by cells of the adventitia and media to meet needs of the host vessel. This growth occurs normally when the vessel wall thickens, but is also stimulated by inflammation or atherosclerosis. MicroCT studies have also shown that a proliferation of the vasa vasorum occurs at sites on the arterial wall associated with atherosclerotic plaques [10]. The prevalence of plaques is greater in patients with diabetes, hypertension, high LDL, and older individuals.

## 4. Insulin, Insulin Resistance, and the Microvasculature

It is interesting to reflect on what is known about the impact of insulin and insulin resistance on microvascular perfusion in other tissues, as this might provide a path for future investigation of vasa vasorum function and dysfunction. Over the past three decades, multiple studies have demonstrated that insulin is a vasoactive hormone [11,12]. In particular, in health insulin enhances microvascular perfusion in skeletal [13] and cardiac muscle [14], adipose [15], skin [16], brain [17,18], and likely other tissues. Insulin acts directly on the vascular endothelial cell insulin receptors [19] to activate the phosphatidylinositol-3-kinase (PI3K) signaling cascade [20] which enhances the activity of endothelial nitric oxide synthase (eNOS). The nitric oxide (NO) formed diffuses to adjacent SMCs in terminal arterioles, where it activates guanylate cyclases leading to SMC relaxation. In addition to, and likely in part due to, its vasorelaxation effect, NO has anti-atherogenic properties [21] even in the setting of diabetes. In addition to insulin, shear stress, and acetylcholine, NO release in the microvasculature of muscle is also provoked by other glucoregulatory hormones (e.g., glucagon-like peptide 1 (GLP-1) [22], adiponectin [23]), and regulates vasoactivity [24,25]. Interestingly, GLP-1 and adiponectin activate eNOS via pathways regulated by cyclic AMP-dependent and AMP- dependent kinases, respectively, which are distinct from insulin’s signaling pathway via PI3K to eNOS.

Conversely, individuals with manifestations of metabolic insulin resistance (patients with metabolic syndrome, hypertension, obesity, type 1 or 2 diabetes) have diminished endothelial responses to shear stress and cholinergic stimuli. Insulin resistant individuals also demonstrate resistance to insulin’s microvascular vasodilatory action and may not only fail to dilate but actually constrict in response to insulin treatment [26]. This is attributable to selective impairment of insulin’s action to increase NO production while a second direct action of insulin to enhance the production of the vasoconstrictor endothelin-1 persists unabated [27,28]. The latter effect is also mediated by insulin binding to its receptor, and downstream activation of the MAP kinase pathway leading to increased transcription of endothelin-1 mRNA. The latter also occurs in the absence of insulin resistance, but in that setting the increase in NO dominates. With varying degrees of insulin resistance there is incremental shifting in the balance between insulin’s vasodilator and vasoconstrictor actions [28]. Both the vasodilator and vasoconstrictor actions of insulin on the vasculature occur quite rapidly. The former within 10–15 min [29], the latter within 1 to 2 h [26]. These microvascular effects of insulin have significant downstream effects on the tissue hosting the microvascular bed. For example, in skeletal muscle inhibiting nitric oxide production in response to insulin diminishes microvascular perfusion and decreases insulin-stimulated glucose uptake [30], i.e., contributes to metabolic insulin resistance. It is of interest that GLP-1 and adiponectin each retain microvascular vasodilatory actions in insulin resistant animals, suggesting that these agents may provide avenues for interventions to prevent microvascular insulin resistance. 

Combined muscle micro-vascular plus metabolic insulin resistance can be provoked experimentally in otherwise healthy laboratory animals by several days of high-fat diet feeding [31], or in humans by raising circulating concentrations of free fatty acids via triglyceride emulsion plus heparin infusion for only a few hours [32]. The deleterious effect of these short exposures to a nutrient overload, perhaps mediated by increased oxidative stress, disrupts vascular endothelial cell regulation. Indeed, multiple studies have reported that in humans a single high-fat meal is sufficient to decrease endothelial function measured by flow mediated dilation [33]. 

To our knowledge, there is no information relating to whether the vasa vasorum of larger elastic or muscular arteries in healthy humans also responds acutely to insulin to increase perfusion of the arterial wall. Nor is there evidence for any impairment in the vasa vasorum of that process by chronic nutrient excess, in a manner comparable to what has been seen in skeletal and cardiac muscle [34], adipose tissue [35,36], and skin [37]. However, it is important to recall that any impairment in vasa vasorum perfusion will most sensitively impact the inner two thirds of the tunica media where the vessels of the vasa vasorum are sparse, and difficult to examine non-invasively [38].

## 5. Experimental Approaches to Studying Vasa Vasorum Structure and Function

As might be expected, progress in our understanding of the role of vasa vasorum in normal physiology as well as in the evolution of atherosclerotic plaque and ASCVD in the clinical setting has been limited by the lack of experimental tools with sufficient resolution to either directly image vasa vasorum or otherwise test its functional behavior. As a result, data analogous to that described above for skin, adipose, cardiac, and skeletal muscle microvasculature in insulin resistant states does not exist for the vasa vasorum. This may be slowly changing as several methods that have been fruitfully used in microvascular studies in other tissues are now being focused on vasa vasorum. 

One example, for over a decade investigators have examined whether contrast ultrasound, which has been productively applied to measure cardiac, skeletal muscle, and adipose microvascular function, could likewise be applied non-invasively to vessel wall microvasculature [39]. These studies typically have focused on the carotid arteries which are large, relatively superficial, and often affected by clinically important atherosclerosis [40]. There have been multiple reports associating increased vessel wall contrast signal intensity (indicative of increased vasa vasorum mass i.e., neovascularization) with luminal plaque lesion [41]. To date none of these studies have provided new insights into the regulation of vasa vasorum function or indeed whether the vasa vasorum is dysfunctional. Virtually all of these studies have used bolus injections of microbubble contrast agents to enhance video intensity in the arterial wall relative to background, while being cautious to avoid shadowing artifacts introduced by the large vessel luminal contrast agent [42]. This differs from the steady state continuous microbubble infusion protocols used for muscle or adipose microvascular imaging where the signal is collected over a larger tissue volume. However, presently, contrast-enhanced ultrasound has, as an anatomic modality, demonstrated that vasa vasorum volume is increased in human type 1 diabetes subjects with or without microvascular complications in the eyes or kidneys [43]. This is consistent with the post mortem anatomic pathologic finding in humans with longstanding type 1 diabetes where expanded but disorganized vasa vasorum is seen. Intriguingly, a recent autopsy histopathologic study of vasa vasorum in coronary arteries of persons with pre-diabetes demonstrated diminished vasa vasorum [44], consistent with vessel rarefaction as is seen in studies of early stages of diabetic retinopathy, and also in muscle tissue in persons with hypertension, obesity, or early diabetes, i.e., persons with insulin resistance. These finding are consistent with a sequence of early onset vascular dysfunction leading to capillary dropout with resulting relative tissue hypoxemia (particularly in the inner third of the tunica media). This could then trigger an angiogenic response including ingrowth of new microvessels from the adventitia (Figure 2). Missing from this attractive construct are any data reporting on the functional capabilities of the vasa vasorum early in the course of vascular stress from either hypertension, hyperglycemia, increased lipids, or local/systemic inflammatory mediators.

Another technique being brought to bear on this knowledge gap is intravascular optical coherence tomography (ivOCT). The OCT method has been extensively applied to study the retinal microvasculature and is now widely used clinically in patients with diabetic retinopathy and other retinal disorders. It has been adapted for use with intravenously placed ultrasound probes that allow their use in the carotid or coronary circulation. It has a higher resolution than contrast ultrasound methods. As an example of its use, Choi and colleagues [45] in a careful study compared segmental endothelial dysfunction (vasodilator response to intracoronary acetylcholine infusion) with ivOCT observed abnormalities in the wall of epicardial coronary arteries. Specifically, they identified signals related to macrophage infiltration and “microchannels”.

## 6. Aberrant Vaso Vasorum Growth in Atherosclerosis

While the regulation of vasa vasorum angiogenesis during normal growth and development has been only lightly studied, in the past several decades extensive study has been conducted of pathologic vasa vasorum angiogenesis during atherogenesis, recognizing it as an integral component of the development of atherosclerotic plaques [46,47]. In that setting, inflammation, vessel remodeling with expansion of the vessel wall and local hypoxemia, as well as increased production of growth factors including VEGF, fibroblast growth factor 2, placental growth factor, TNFα and others occurs in areas of developing atheroma. This provokes/promotes greater vascular permeability, and increased vascular adhesion molecule expression. This, in turn, increases the influx of apolipoproteins, leukocytes, macrophages, each of which contribute to advancing atheroma progression. The new vessels in the expanding vasa vasorum form a complex web that can progress into the core of the plaque within the intima. There, vessel leakage and hemorrhage may contribute to plaque instability and rupture. Indeed, improved understanding of vasa vasorum angiogenesis has led to a greater appreciation of the importance of the adventitial vasa vasorum in particular. Some important observations include that: (a) the adventitial vasa vasorum is more abundant at sites of luminal atherosclerotic plaque [48]; (b) inflammatory cells (macrophages, B and T lymphocytes) are more abundant in the adventitia than in the media or intima of the vessel wall; (c) injury of the parent vessel endothelium first provokes migration of inflammatory cells into the adventitia [49]; (d) in experimental atherosclerotic animal models, neovascularization begins within a few weeks of feeding animals a hypercholesterolemic diet [50], perhaps preceding endothelial dysfunction; and (e) pharmacologic inhibition of vasa vasorum neovascularization diminishes atherosclerosis progression [46]. 

## 7. Microvascular Dysfunction, Insulin Resistance and Accelerated Atherosclerosis: Is Vasa Vasorum a Missing Link? 

As noted previously, metabolic insulin resistance, as found in subjects with metabolic syndrome, type 2 diabetes, hypertension, or obesity, significantly increases atherosclerotic vascular disease risk. Metabolic insulin resistance is strongly associated with microvascular insulin resistance (typically measured by a diminished action of insulin to enhance perfusion) in a number of tissues. Importantly, it is not known whether microvascular insulin resistance extends to the vasa vasorum in the wall of major vessels or whether some of the same cellular changes that provoke microvascular insulin resistance also provoke vascular changes associated with early stages of atherogenesis (Figure 2). 

Indications that this may be the case include: (a) insulin’s action to enhance tissue perfusion is strongly linked to it increasing NO availability and NO [11,51] is widely regarded to have athero-protective actions [21]; (b) vascular insulin resistance is associated with increases of ROS production [52] and adhesion molecule expression by the endothelium, both of which are seen in early stages of atherosclerotic plaque development; and (c) nutrient overload promotes migration of leukocytes into the wall of blood vessels and into body fat depots [53], early changes in both vascular insulin resistance and atherosclerosis. As atherosclerosis development proceeds the arterial wall thickens, particularly the intima, secondary to fibroblast and SMC proliferation and migration from the media to the intima. This vessel growth and wall thickening stimulates vasa vasorum expansion (as in normal growth) to avoid tissue hypoxia. However, in the inflammatory milieu of the atheroma, factors like TNFα, VEGF, and others impair vessel maturation resulting in more “leaky” vessels that contribute to a spiraling upward of inflammatory cell and lipid infiltration into the developing plaque. This aberrant neovascularization is closely linked to an expanding influx of inflammatory cells that is seen in the adjacent adventitial vasa vasorum (aided by increased expression of adhesion molecules including e-selectin, ICAM, and VCAM) [54]. The vessels of this neovascular vasa vasorum are small and fragile and tend to rupture easily. When this occurs in the intima it can lead to intraplaque hemorrhages, a major driver of thrombus formation and cardiovascular events. 

## 8. Research Challenges: Knowable Unknowns

Motivation for this review arose from the recognition of an expanding body of evidence demonstrating prevalent microvascular insulin resistance in persons with known metabolic insulin resistance together with an even more rapidly growing body of work implicating a central role for vasa vasorum in atherogenesis. If early vasa vasorum dysfunction presages atherosclerotic plaque initiation and progression, then normal vasa vasorum function and early dysfunction deserves more investigation. The paucity of information on vasa vasorum function is in part attributable to its small size and general inaccessibility. Currently we lack imaging resolution that allows in vivo measurement of vasa vasorum responses to acute interventions in the presence and absence of insulin resistance. Post mortem Micro-CT studies have demonstrated major changes in vasa vasorum structure in settings of established atherosclerotic stenotic injury. These studies are demanding, and best adapted to visualization of vessels (e.g., coronary or carotid arteries) many days or weeks after injury has occurred. Heistad and colleagues, in classic studies, nicely demonstrated the utility of radiolabeled microspheres for assessing acute changes in flow through the vasa vasorum in vivo [6,7]. Such techniques have to our knowledge not been applied to studies of vasa vasorum function in response to acute changes of nutrient or hormonal milieu, as has been done for studies of endothelial function or microvascular perfusion in other tissues. Such functional studies would appear highly desirable and should be accompanied by measurements of gene expression within specific cellular components of the parent vessel wall as well as the vascular elements of the vasa vasorum. Combined information derived from such studies should help define early changes in both the parent vessel wall and the vasa vasorum that could identify pathways for intervention very early in atherosclerosis development before there are identifiable morphologic lesions.

## Figures and Tables

**Figure 1 ijms-22-07574-f001:**
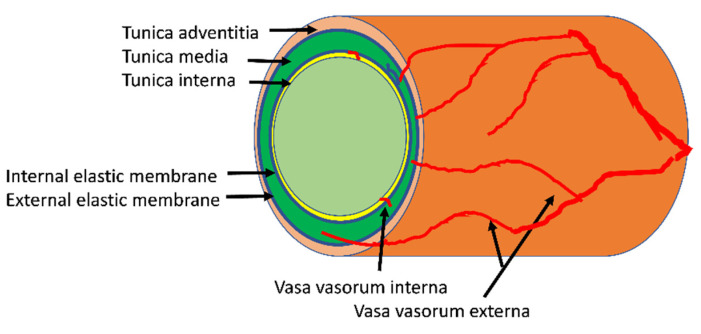
Most microvessels of the vasa vasorum enter the vessel wall from the adventitial layer where they tend to course longitudinally along the vessel axis. After penetrating to the media of the arterial wall they more typically orient circumferentially through the media. Much less abundant is the vasa vasorum interna with its short vessels feeding small areas of the innermost third of the vessel wall.

**Figure 2 ijms-22-07574-f002:**
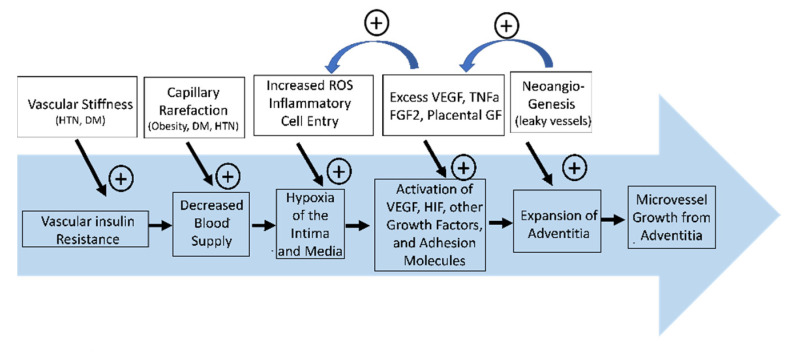
Vascular insulin resistance from metabolic disease such as hypertension and diabetes causes thickening of the intimal layer of the vasa vasorum. This (along with capillary rarefaction) leads to a decrease in blood flow feeding the arterial wall and relative hypoxia of the intimal and medial layers. This activates VEGF, HIF1, HIF2, TNFa, fibroblast growth factor 2, placental growth factor, and adhesion molecules and initiates a positive feedback loop that promotes more hypoxia as these inflammatory molecules increase ROS. This leads to expansion of the adventitia and promotes neoangengiogensis and further vessel injury.

## Data Availability

Not applicable.

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
