# Peer review of "Early Microvascular Dysfunction: Is the Vasa Vasorum a “Missing Link” in Insulin Resistance and Atherosclerosis"

_ijms, 2021, doi:10.3390/ijms22147574_

Round 1
Reviewer 1 Report
Interesting, well written and up-to-date review.
The order of the paragraphs should be modified according to a more logical seuqence:
paragraphs "5.Normal growth of the vasa vasorum" and "6.Aberrant vasa vasorum growth in atherosclerosis" should run after the paragraph 2.Structure and function.
Lines 145-147, there is the need for a bibliographic reference.
Author Response
Comment 1: Interesting, well written and up-to-date review.
Response 1: We thank the referee for his/her positive assessment of the review.
Comment 2: The order of the paragraphs should be modified according to a more logical sequence: paragraphs "5.Normal growth of the vasa vasorum" and "6.Aberrant vasa vasorum growth in atherosclerosis" should run after the paragraph 2.Structure and function.
Response 2: We greatly appreciate the suggestion and have partially implemented. We have moved section 5 Normal growth of the vasa vasorum to follow section 2, as the reviewer suggested. We have not however followed that with section 6 as he/she suggested. We felt it important to review first the normal physiology of insulin action on the microvasculature before proceeding to pathologic changes that occur in atherosclerosis. We very much appreciate the reviewer’s suggestion, and hope that he/she can agree with our rationale for this
Comment 3: Lines 145-147, there is the need for a bibliographic reference. Response 3: our apologies, we could not identify the statement that required referencing, please clarify.
Reviewer 2 Report
I appreciate the authors for the concise and clear review of the topic. I would like to suggest few comments as below,
- Would suggest the authors to list a table of studies with atherosclerotic animal models, their effects on vasa vasorum and proposed associated mechanism to improve readability.
-
Diabetic patients who are insulin resistant are less susceptible to aortic aneurysm which is an athero-thrombotic condition. Can the authors comment on this discrepancy compared to atherosclerosis in carotid artery? How does vasa vasorum play into this condition which also has microvascular dysfunction?
Minor comments:
- Line 23: Expand the abbreviation at first use
- Line 98: Reference error
Author Response
Comment 1: I appreciate the authors for the concise and clear review of the topic. I would like to suggest few comments as below.
Response 1: we thank reviewer 2 for his/her positive comment.
Comment 2: Would suggest the authors to list a table of studies with atherosclerotic animal models, their effects on vasa vasorum and proposed associated mechanism to improve readability.
Response 2: we appreciate the suggestion, and considered it carefully. However, while there are numerous studies that have commented on the role of the vasa vasorum in atherosclerotic animal models, (particularly genetically modified mice) none have defined and associated mechanism in an unambiguous fashion. This is particularly complicated by the uncertain relationship between the atherosclerosis developed in rodent models and that in larger animals where there is a pre-existing vasa vasorum not simply one provoked by atherogenesis. Consequently, explaining this complexity in the table seems beyond the scope of what we can accomplish in this brief review.
Comment 3: Diabetic patients who are insulin resistant are less susceptible to aortic aneurysm which is an athero-thrombotic condition. Can the authors comment on this discrepancy compared to atherosclerosis in carotid artery? How does vasa vasorum play into this condition which also has microvascular dysfunction?
Any comment we might make relating to the observation that persons with diabetes and insulin resistance are less prone to development of aortic aneurysm but greater carotid atherosclerosis would be purely speculative. To our knowledge, there is no information extant identifying a differential structural or functional role for vasa vasorum for this long-standing clinical observation. Consequently, we don’t feel addressing this issue here would provide any insight to the reader.
Minor comments:
Line 23: Expand the abbreviation at first use -Response: we have expanded the abbreviation for cardio metabolic disease in the introduction.
Line 98: Reference error Corrected, thank you